# Numerical Simulation of Non-Stationary Parameter Creep Large Deformation Mechanism of Deep Soft Rock Tunnel

**Jiancong Xu** [1,*] **, Haiyang Wen** [1] **, Chen Sun** [1] **, Chengbin Yang** [1] **and Guorong Rui** [2]

1    Department of Geotechnical Engineering, Tongji University, Shanghai 200092, China;
     1410258@tongji.edu.cn (H.W.); 1832539@tongji.edu.cn (C.S.); 1932507@tongji.edu.cn (C.Y.)
2    China Railway 20th Bureau Group Co., Ltd., Xi'an 710016, China; sycsdxmb@163.com
*    Correspondence: xjc1008@tongji.edu.cn

**Abstract:** The accelerated creep plays an important role in the disasters of soft-rock tunnels under high stress. However, most of previous studies only involved attenuation creep and uniform creep. Large deformation disasters of soft rock occurred during the tunneling process in the Qianzhou–Sanyangchuan Tunnel, Gusu Province, China. In the paper, we developed the nonlinear generalized Nishihara rheological model with non-stationary parameter creep (NGNRM) to simulate the accelerated creep behaviors of soft rocks under high stress, and implemented it in ABAQUS, to reveal the mechanism of large deformation of soft rock. We proposed the multi-objective back analysis method of surrounding rock mechanical parameters based on the eXtreme Gradient Boosting and the non-dominated sorting genetic algorithm-II. In addition, the orthogonal test design method was used to determine the main parameters affecting the displacement of the tunnel. Using the proposed method, we can evaluate the large deformation mechanism of deep soft rock tunnels, and scientifically determine when to reinforce to prevent a large deformation disaster of the tunnel.

**Keywords:** soft rock; creep; large deformation; numerical simulation; XGBoost; NSGA-II

## 1. Introduction

In soft rock strata, the large squeezing deformation of surrounding rock often occurs during tunneling under high stress [1–6]. Many interesting cases of tunnels, for example, the section of the F7 fault fracture zone of the Wushaoling Railway Tunnel [7], and the Saint Martin La Porte access adit of the Lyon–Turin Base Tunnel [2,3], were recorded as representative of large squeezing deformation disasters. These disasters are mostly caused by soft rock rheology [1–4]. Therefore, it is of theoretical and practical interest to investigate the large deformation mechanism of deep soft rock tunnels and to explore an efficient numerical modeling method.

The accelerated creep of soft rock plays an important role in tunnel large deformation disasters under high stress. Many previous works were devoted to investigating the time-dependent deformation of surrounding rock around tunnels using viscoelastic models and viscoplastic models [5]. However, these studies only involved the attenuation creep and uniform creep of soft rocks [1–6], and seldom involved the accelerated creep.

The identification of rheological curves in the high-stress soft rock test shows that the viscosity coefficient related to the non-stationary parameter creep is often a function of the stress level [8], and the previous viscoelastic–plastic models such as the Nishihara model, cannot simulate the accelerated rheological characteristics of soft rock. Therefore, only the nonlinear rheological model with the non-stationary parameter creep can completely present the accelerated creep characteristics of soft rock under high stress.

In engineering practice, with the variation of the engineering environment and tunneling process, the mechanical parameters of surrounding rock usually have uncertainty. The deformation of surrounding rock and the supporting structure is the most direct, obvious

and comprehensive macroscopic manifestation on the force change of the surrounding rock support system in the time–space domain of tunneling and support. Therefore, the mechanical parameters obtained by the displacement back analysis can reflect the influence of geo-stress, the construction method, and the tunnel structure parameters as a whole. At present, the back analysis has been widely used to obtain the mechanical parameters of surrounding rocks by simulating their complicated mechanical behaviors through simple models combined with measured data [9–15].

Kavanagh and Clough [16] proposed the back analysis theory for the first time, and they inverted the elastic modulus of elastic solids based on the finite element method. Since then, some traditional optimization algorithms such as Levenber–Marquard, Gauss–Newton, Bayesian and other methods were applied to the field of back analysis. Gioda [17] used the mixed algorithm of the variable rotation method and the simplex method to invert the mechanics of elastic–plastic rock mass. Later, some scholars proposed the parameter inverted methods for five commonly used creep models (such as Maxwell, Kelvin, generalized Kelvin and Burgers models) [18,19].

Although the above methods have made great progress in the field of back analysis of geotechnical parameters, with the progress of machine learning theory, some scholars have proposed numerous algorithms with higher computational accuracy and computational efficiency. Compared with traditional algorithms, these algorithms can better deal with the complex nonlinear relationship in the geotechnical back analysis, so as to achieve higher accuracy. The mechanical parameters of surrounding rock may be obtained by the displacement back analysis using neural networks and machine learning, such as the genetic algorithm and particle swarm optimization [20–27].

In this study, we propose a nonlinear generalized Nishihara rheological model with the non-stationary parameter creep (NGNRM), and implement it in ABAQUS. Furthermore, we propose the multi-objective back analysis method (MBAM) of surrounding rock mechanical parameters using the eXtreme Gradient Boosting (XGBoost) and the non-dominated sorting genetic algorithm-II (NSGA-II) for deep soft rock tunnels, to invert the multiple mechanical parameters of surrounding rock at the same time. Finally, according to the displacement data of the completed sections or a period of time of the inversion section, we use the proposed XGBoost–NSGA-II-based MBAM to identify the crucial rheological parameters of soft rocks, and use the proposed NGNRM method to simulate and analyze the whole process of large deformation disaster evolution of a deep soft rock tunnel, and reveal its non-stationary parameter creep large deformation mechanism. These methods have been successfully applied to the Qianzhou–Sanyangchuan Tunnel located in Gusu Province, China, with higher precision.

The novelty of this paper is that we propose the NGNRM to fully express the accelerated creep characteristics of soft rock under high stress, and the XGBoost–NSGA-II-based MBAM, to help in the understanding of numerical simulation rheological models in large creep evolution during soft rock tunneling.

## 2. NGNRM with Non-Stationary Parameter Creep

The accelerated creep of soft rock plays an important role in the large deformation disasters of tunnels. From the introduction, it is known that the previous studies seldom reflect the accelerated creep in the whole process of soft rock large deformation. In this paper, we used the NGNRM (see Figure 1) representing the accelerated creep characteristics of soft rocks, to investigate the large deformation mechanism of soft rocks. The NGNRM consists of a Hooke body, a Kelvin body, and a viscoplastic body with the viscosity coefficient as a function of stress.

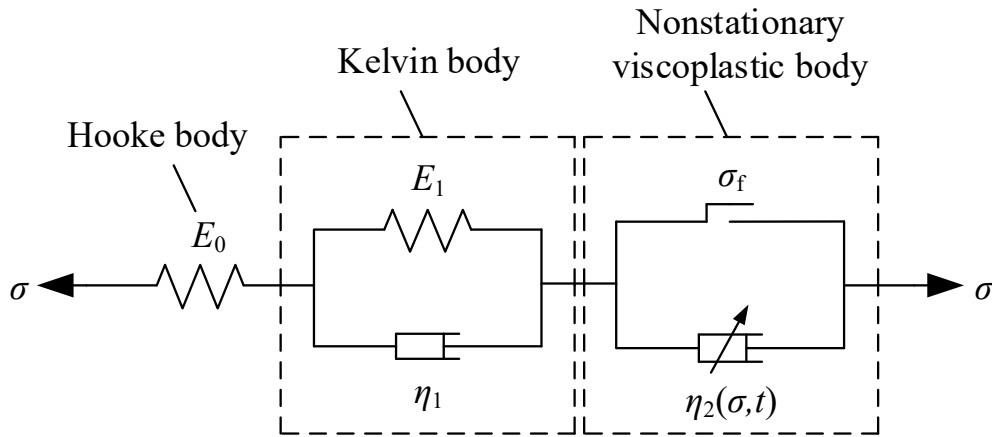

**Figure 1.** Rheological model with variable viscoplastic viscosity coefficient.

In Figure 1, $E_0$ is the elastic modulus of a Hooke body; $E_1$ and $\eta_1$ are the elastic modulus of a Hooke body and the viscosity coefficient of a Newton body in a Kelvin body, respectively; $\sigma_f$ and $\eta_2$ are the yield stress of a St. Venant body and the viscosity coefficient of a dashpot in the non-stationary viscoplastic body, respectively, and $\eta_2$ is as a function of stress $\sigma$ and time $t$.

When $\sigma \geq \sigma_f$, for the nonlinear Bingham body, the expression of the viscosity coefficient $\eta_2$ is denoted by Equation (1) [28].

$$\begin{cases} \varepsilon = \dfrac{\sigma}{E_0} + \dfrac{A(\sigma - \sigma_f)t}{E_0\left[T + t\left(1 - \frac{\sigma - \sigma_f}{\sigma_s}A\right)\right]} \\ \eta_2(\sigma) = \dfrac{E_0}{AT}\left[T + t\left(1 - \frac{\sigma - \sigma_f}{\sigma_s} \cdot A\right)\right]^2 \end{cases} \tag{1}$$

where $\sigma$ is the current stress; $\sigma_s$ is the long-term strength of a Bingham body; $A$ is a dimensionless parameter; $T$ is the time for the surrounding rock to yield; $t$ is load time.

It is noted from Equation (1) that when $\sigma = \sigma_f + \sigma_s/A$, nonlinear creep degenerates into linear creep, $\eta = E_0T/A$, and strain increases at the constant rate $\dot{\varepsilon} = \sigma_s/E_0T$, that is, steady creep; when $\sigma > \sigma_f + \sigma_s/A$, $\eta_2$ gradually decreases with $t$, reflecting the accelerated creep process of soft rock in a high-stress state.

Therefore, when $\sigma \geq \sigma_f$, the one-dimensional creep equation of the nonlinear viscoelastic–plastic model is denoted by Equation (2).

$$\varepsilon(t) = \sigma\left\{\frac{1}{E_0} + \frac{1}{E_1}\left[1 - \exp\left(-\frac{E_1}{\eta_1}t\right)\right]\right\} + \frac{A(\sigma - \sigma_f)t}{E_0\left[T + t\left(1 - \frac{\sigma - \sigma_f}{\sigma_s}A\right)\right]} \tag{2}$$

In this paper, the initial strain method is used to calculate the viscoelastic–plastic constitutive equations of the NGNRM, and its ABAQUS user material subprogram is developed by FORTRAN, which is embedded in the ABAQUS. The flowchart of the user-defined material mechanical behavior (UMAT) subroutine development for the NGNRM is shown in Figure 2.

UMAT is a FORTRAN program interface provided by ABAQUS to users to define their own material properties, enabling users to use material models that are not defined in the ABAQUS material library. The UMAT realizes the data exchange with ABAQUS through the interface with the ABAQUS main solver.

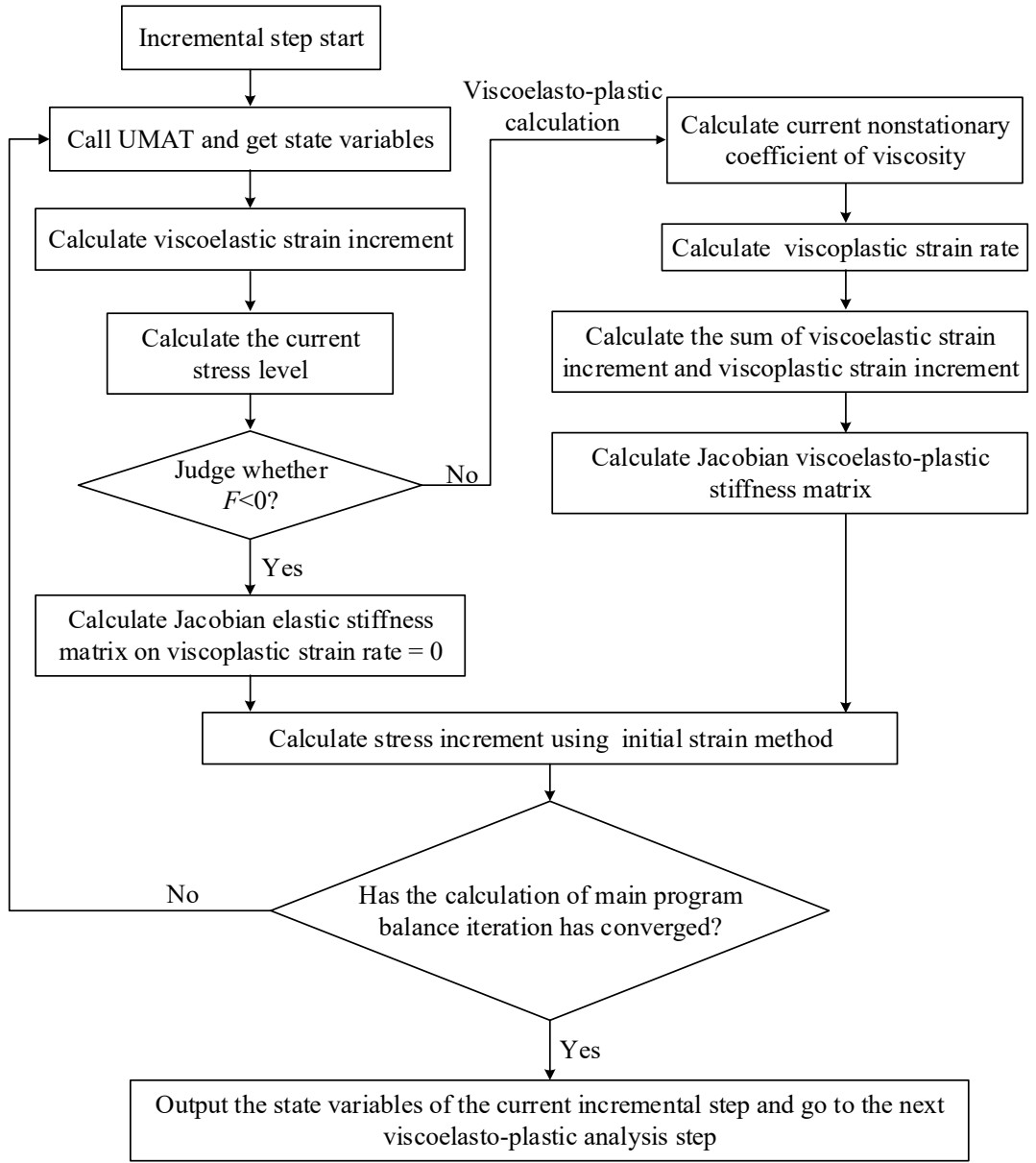

**Figure 2.** UMAT flowchart of nonlinear generalized Nishihara rheological analysis.

Drucker–Prager yield criterion is adopted in this paper, denoted by Equation (3).

$$
\begin{cases}
F(\sigma, c) = \sqrt{J_2} + \eta I_1 - \xi c \\
\eta = \dfrac{\sqrt{3}\sin\varphi}{3\sqrt{(3+\sin^2\varphi)}} \\
\xi = \dfrac{\sqrt{3}\cdot\cos\varphi}{\sqrt{3}\sqrt{(3+\sin^2\varphi)}}
\end{cases}
\tag{3}
$$

where $c$ and $\varphi$ are the internal cohesion and internal friction angle, respectively; $I_1$ is the first invariant of stress; $J_2$ is the second invariant of deviatoric stress.

The UMAT subroutine of the NGNRM are implemented by following steps:

*Step 1* Call the UMAT subroutine. ABAQUS provides state variables such as stress and strain at the previous moment and the increment of strain at the current moment.

*Step 2* Calculate the viscoelastic strain increments of elements.

*Step 3* Calculate the current stress level and the yield function of the viscoplastic body, and judge whether <0. If yes, it will enter the next step, and conduct the viscoelastic calculation. Otherwise, it will enter Step 5.

*Step 4* Calculate the Jacobian elastic stiffness matrix when the viscoplastic strain rate = 0.

*Step 5* Conduct the viscoelastic–plastic calculation. Calculate the nonstationary coefficient of the viscosity and viscoplastic strain rate of elements.

*Step 6* Calculate the sum of viscoelastic strain increment and viscoplastic strain increment of elements.

*Step 7* Calculate the Jacobian viscoelastic–plastic stiffness matrix.

*Step 8* Calculating stress increment using the initial strain method.

*Step 9* Determine whether the balance iteration calculation of the main program has converged. If yes, it will enter the next step. Otherwise, it will return to Step 1.

*Step 10* Output the state variables of the current incremental step and go to the next viscoelastic–plastic analysis step.

## 3. XGBoost

XGBoost [29,30] takes into account the second order term of Taylor expansion and introduces richer derivative information, so that it can better capture nonlinear information. Besides, XGBoost can reduce the over-fitting effect that may be caused by the gradient boosting algorithm by introducing regular terms.

XGBoost is an improvement on the gradient boosting algorithm, developed by Chen and Guestrin [29]. It uses Newton's method to solve the extreme value of the loss function, and adds a regularization term to the loss function. Its loss function is denoted by the second-order Taylor expansion. On training, the objective function is composed of two parts: the first part is the loss of the gradient boosting algorithm, and the second part is the regularization term.

Given a training sample data set $T = \{(x_1, y_1), (x_2, y_2), \cdots, (x_N, y_N)\}$, $x_i \in \chi \in R^n$, $\chi$ is the input space of samples, $y_i \in Y \in R$, $Y$ is the output space, $n$ is the dimension of input space. If the input space $\chi$ is divided into non-intersecting regions $R_1, R_2, \cdots, R_J$, and the constant output $c_j$ is determined on each region, then the tree can be denoted by Equation (4).

$$T(x; \Theta_k) = \sum_{j=1}^{J} c_j I(x \in R_j) \tag{4}$$

where $\Theta = \{(R_1, c_1), (R_2, c_2), \cdots, (R_J, c_J)\}$ represents the region division of the tree and the constants on each region; $J$ is the complexity of the regression tree, that is, the number of leaf nodes.

XGBoost's boosting tree model can be expressed by Equation (5) as an additive model of decision trees.

$$f_K(x) = \sum_{k=1}^{K} T(x; \Theta_k) \tag{5}$$

where $T(x; \Theta_k)$ is the decision tree; $\Theta_k$ is parameters of the decision tree; $K$ is the number of trees.

The boosting tree algorithm in XGBoost uses a forward step-by-step algorithm as follows:

Firstly, the initial boosting tree $f_0(x) = 0$ is determined.

Secondly, the model at the *k*-th step is denoted by Equation (6).

$$f_k(x) = f_{k-1}(x) + T(x; \Theta_k), \ k = 1, 2, \cdots, K \tag{6}$$

where $f_{k-1}(x)$ is the current model; $\Theta_k$ is the parameters of the decision tree, determined by the minimization of empirical risk, denoted by Equation (7).

$$\hat{\Theta}_k = \underset{\Theta_k}{\operatorname{argmin}} \sum_{i=1}^{N} L(y_i, f_{k-1}(x_i) + T(x_i; \Theta_k)) \tag{7}$$

where $\hat{\Theta}_k$ is parameters of the $k$-th tree; $L(\boldsymbol{y}_i, f_{k-1}(\boldsymbol{x}_i) + T(\boldsymbol{x}_i; \Theta_k))$ is the loss value between the observed value $y$ and the predicted value $f_k(\boldsymbol{x})$ for the $i$-th sample.

The loss function may be defined as $L(\boldsymbol{y}, f(\boldsymbol{x})) = (\boldsymbol{y} - f(\boldsymbol{x}))^2$ using the least square error criterion, denoted by Equation (8).

$$l(\boldsymbol{y}, \boldsymbol{y}') = L(\boldsymbol{y}, f_{k-1}(\boldsymbol{x}) + T(\boldsymbol{x}; \Theta_k)) = [\boldsymbol{y} - f_{k-1}(\boldsymbol{x}) - T(\boldsymbol{x}; \Theta_k)]^2 = [\boldsymbol{r} - T(\boldsymbol{x}; \Theta_k)]^2 \quad (8)$$

where $\boldsymbol{r} = \boldsymbol{y} - f_{k-1}(\boldsymbol{x})$ is the fitting residual of the current model; $f(\boldsymbol{x})$ is the prediction value obtained by the fitting model, $\boldsymbol{y}' = f(\boldsymbol{x})$; $f_{k-1}(\boldsymbol{x})$ is the fitting model obtained at the $k-1$-th step.

Thirdly, calculate the residual by Equation (9).

$$\boldsymbol{r}_{k_i} = \boldsymbol{y}_i - f_{k-1}(\boldsymbol{x}_i), \ i = 1, 2, \cdots, N \quad (9)$$

Learn a regression tree based on the fitting residual $\boldsymbol{r}_{k_i}$, and obtain $T(\boldsymbol{x}; \Theta_k)$.
Fourthly, update $f_k(\boldsymbol{x}) = f_{k-1}(\boldsymbol{x}) + T(\boldsymbol{x}; \Theta_k)$.

Fifthly, obtain boosting tree of regression problem $f_K(\boldsymbol{x}) = \sum\limits_{k=1}^{K} T(\boldsymbol{x}; \Theta_k)$.

The loss function for XGBoost is defined as $L(\phi)$, denoted by Equation (10).

$$L(\phi) = \sum_{i=1}^{n} l(\boldsymbol{y}_i, \boldsymbol{y}'_i) + \sum_k \boldsymbol{\Omega}(f_k) = l(\boldsymbol{y}, \boldsymbol{y}') + \sum_k \boldsymbol{\Omega}(f_k) = [\boldsymbol{r} - T(\boldsymbol{x}; \Theta_k)]^2 + \sum_k \boldsymbol{\Omega}(f_k) \quad (10)$$

where $n$ is the number of training samples; $\sum\limits_{i=1}^{n} l(y_i, y'_i)$ is the loss value caused using the gradient boosting algorithm, calculated by Equation (8), assuming it is a differentiable convex function; $\boldsymbol{\Omega}(f_k)$ is the regularization term; $f_k$ is the weak learner function.

The complexity of the XGBoost model is defined by the regularization term $\boldsymbol{\Omega}(f_k)$, denoted by Equation (11).

$$\boldsymbol{\Omega}(f_k) = \gamma T_{\ln} + \frac{1}{2}\lambda \|\boldsymbol{\omega}\|^2 = \gamma T_{\ln} + \frac{\lambda}{2}\sum_{j=1}^{J} \omega_{tj}^2 \quad (11)$$

where $\gamma$ and $\lambda$ are two artificially set coefficients; $\boldsymbol{\omega}$ is the vector formed by the values of all leaf nodes of the decision tree; $T_{\ln}$ is the number of leaf nodes; $\omega_{tj}$ is the weight of leaf $j$.

The ultimate goal of XGBoost is to minimize the objective function in Equation (10). The objective function in Equation (10) is approximately solved by the Newton's method, and denoted by the second-order Taylor expansion at the point $y'_{i,t-1}$ and Equation (12).

$$L_t(\phi) \approx \sum_{i=1}^{n} \left[ l\left(y_i, y'_{i,t-1}\right) + \frac{\partial L(y_i, y'_i)}{\partial y'_i}\bigg|_{y'_i = y'_{i,t-1}} f_t(x_i) + \frac{1}{2}\frac{\partial^2 L(y_i, y'_i)}{\partial y'^2_i}\bigg|_{y'_i = y'_{i,t-1}} f^2_{t-1}(x_i) \right]$$
$$+\gamma T_{\ln} + \frac{\lambda}{2}\sum_{j=1}^{T} \omega_{tj}^2 \quad (12)$$

where $f_t(x_i)$ is the current weak learner; $x_i$ is the $i$-th training sample.

To simplify the expression of Equation (12), the first-order negative gradient and the second-order negative gradient are recorded as $g_{ti}$ and $h_{ti}$ denoted by Equation (13).

$$\begin{cases} g_{ti} = \dfrac{\partial L(y_i, y'_i)}{\partial y'_i}\bigg|_{y'_i = y'_{i,t-1}}, h_{ti} = \dfrac{\partial^2 L(y_i, y'_i)}{\partial y'^2_i}\bigg|_{y'_i = y'_{i,t-1}} \\ G_t = \sum_{i \in I_j} g_{ti}, H_t = \sum_{i \in I_j} h_{ti} \end{cases} \quad (13)$$

Delete the constant term of Equation (12) that is independent of $f_t(x_i)$. The form of the final loss function can be simplified to Equation (14) when $G_t = \sum_{i \in I_j} g_{ti}$ and $H_t = \sum_{i \in I_j} h_{ti}$.

$$L_t = \sum_{i=1}^{T} \left[ G_t \omega_j + \frac{1}{2}(H_t + \lambda)\omega_j^2 \right] + \gamma T_{\ln} \tag{14}$$

## 4. NSGA-II –XGBoost Based Multi-Objective Back Analysis

A numerical model and the optimization method are two key components of the back analysis. For deep soft rock tunnels, the numerical modeling is time consuming. In this paper, XGBoost was used to represent numerical model and to map the relationship between the mechanical parameters of surrounding rock and the displacement of each measuring point, to reduce the computing time greatly. NSGA-II was adapted to search multi-objectively the geomechanical parameters as an optimization method.

### 4.1. XGBoost-Based Relationship between Displacement and Geomechanical Parameters

We used XGBoost to map the nonlinear relationship between geomechanical parameters such as elastic modulus, internal friction angle, cohesion, geo-stress coefficients, and monitored displacements, where the surrogate model is built to replace the numerical simulation model. The mathematical model of the multi-output XGBoost, $XGBoost(X)$, is defined as:

$$\begin{cases} XGBoost(X): R^N \to R^M \\ Y = XGBoost(X) \end{cases} \tag{15}$$

where $X = (x_1, x_2, \cdots, x_N)$, $x_i$ $(i = 1, 2, \cdots, N)$ is a vector of geomechanical parameters such as elastic modulus, internal friction angle, and cohesive force, etc.; $N$ is the number of mechanical parameters; $Y = (y_1, y_2, \cdots, y_M)$ is the $M$ dimensional vector of the monitoring data, such as displacement. In this case, the observable output is the displacement in the back analysis, correspondingly, $M$ represents the dimension of displacement data.

To obtain $XGBoost(X)$, a training process based on a known dataset is needed. The necessary training samples were created for this work by combining the FEM numerical analysis and test design, which is used to obtain the displacements of the surrounding rock initial support system according to a given set of mechanical parameters of the surrounding rock. For this study, the sampling by the uniform test design method was adopted to build samples for mechanical parameters of surrounding rock, and these geomechanical parameters were defined as the input of XGBoost. The displacement was defined as the output of XGBoost.

### 4.2. XGBoost–NSGA-II-Based Back Analysis

We propose the XGBoost–NSGA-II-based back analysis method for the deep soft rock tunnel engineering. The monitored displacements at different monitoring points are denoted herein as $Y_{\text{monitor1}}, Y_{\text{monitor2}}, \cdots, Y_{\text{monitor}d}$; the predicted displacement by the XGBoost-based surrogate model at different monitoring points are denoted as $y_1, y_2, \cdots, y_d$.

The non-dominated sorting genetic algorithm-II (NSGA-II) is a fast non-dominated multi-objective optimization algorithm based on the Pareto optimal solution with an elite retention strategy [31,32].

The objective function of multi-objective optimization is a set of functions containing multiple members, and its optimization form is denoted by Equation (16).

$$\boldsymbol{\theta}^* = \underset{\boldsymbol{\theta}}{\arg\min} L(\boldsymbol{\theta}) = \underset{\boldsymbol{\theta}}{\arg\min} \{ \ell_1(\boldsymbol{\theta}), \ell_2(\boldsymbol{\theta}), \cdots, \ell_d(\boldsymbol{\theta}) \} \tag{16}$$

where $\boldsymbol{\theta}$ is the geomechanical parameters to be inverted; "argmin" represents the variable $\boldsymbol{\theta}$ when the objective function $L(\boldsymbol{\theta})$ is minimized, i.e., $\boldsymbol{\theta}^*$; $d$ is the number of monitoring points for each section of tunnel.

An objective function is set for each monitoring point, and each objective function is represented by the mean square error (MSE). Finally, the multi-objective function is denoted by Equation (17).

$$
\min L(\boldsymbol{\theta}) = \begin{cases} \min \ell_1 = (Y_{\text{monitor1}} - y_1)^2 \\ \min \ell_2 = (Y_{\text{monitor2}} - y_2)^2 \\ \cdots \\ \min \ell_d = (Y_{\text{monitord}} - y_d)^2 \end{cases} \tag{17}
$$

where $\ell_i$ $(i = 1, 2, \cdots, d)$ represents the MSE of the $i$-th monitoring point.

NSGA-II is used to solve the model parameters according to the following steps:

***Step 1*** Structural representation of solutions. The solution of the model optimization problem may be expressed by a floating point vector or binary vector. The binary vectors are used as chromosomes to represent the true value of the solution space, and the length of the vector is determined by the accuracy of solving the problem. The chromosome number of the initial population is *pop_size*.

***Step 2*** Generation of the initial population of feasible solutions. If there is no experience and prior knowledge, the initial population can be randomly generated.

***Step 3*** The size of the initial population *pop_size*. The number of chromosomes *pop_size* is 50~100. By random sampling for *pop_size* times, the initial sample of the population at $t = 0$ may be obtained.

***Step 4*** Evaluation function. Supposing the chromosomes are $V_1, V_2, \cdots, V_{pop\_size}$, an order-based evaluation function is defined by Equation (18).

$$
\text{eval}(V_i) = \alpha (1 - \alpha)^{i-1} (i = 1, 2, \cdots, pop\_size, \alpha = 0.5) \tag{18}
$$

The *pop_size* chromosomes are decoded, and the adaptability of each chromosome is evaluated. If the optimal solution remains unchanged for no less than three consecutive generations, the solution is the optimal solution, therefore proceed to Step 9 to obtain the optimal solution. Otherwise, continue to Step 5 to select chromosomes.

***Step 5*** Chromosome selection. The selection process of chromosomes is based on spinning the wheel *pop_size* times. Based on the fitness of each chromosome, each rotation selects a chromosome for the new population. The selection process is as follows:

First, for each chromosome $V_i$, calculate the cumulative probability $q_i$ by Equation (19).

$$
\begin{cases} q_0 = 0 \\ q_i = \sum\limits_{j=1}^{i} \text{eval}(V_j) (i = 1, 2, \cdots, pop\_size) \end{cases} \tag{19}
$$

Second, generate a random number $r$ from $\left[0, q_{pop\_size}\right]$. If $q_{i-1} < r \leq q_i$, select the $i$-th chromosome $V_i (1 \leq i \leq pop\_size)$.

Third, repeat the above two steps for the *pop_size* times to get *pop_size* duplicated chromosomes.

***Step 6*** Conduct the hybridization operation to generate *pop_size* new chromosomes with an appropriate crossover probability $P_c = 0.4$~$0.9$.

***Step 7*** Select $P_m \times pop\_size$ chromosomes from *pop_size* chromosomes for mutation operation with an appropriate probability $P_m = 0.001$~$0.1$.

***Step 8*** Let $t = t + 1$, and go to Step 4.

***Step 9*** Solving the model is terminated, and the optimal solution $\theta^*$ is obtained.

### 4.3. Computation Procedure for the XGBoost–NSGA-II-Based Back Analysis

The algorithm for the XGBoost–NSGA-II-based back analysis uses the NSGA-II approach to calculate the optimal combination of geomechanical parameters in the deep soft rock tunnel engineering. XGBoost is adopted to surrogate this relationship instead of numerical models used in traditional back analysis methods. The back analysis procedure is shown in Figure 3 and explained below:

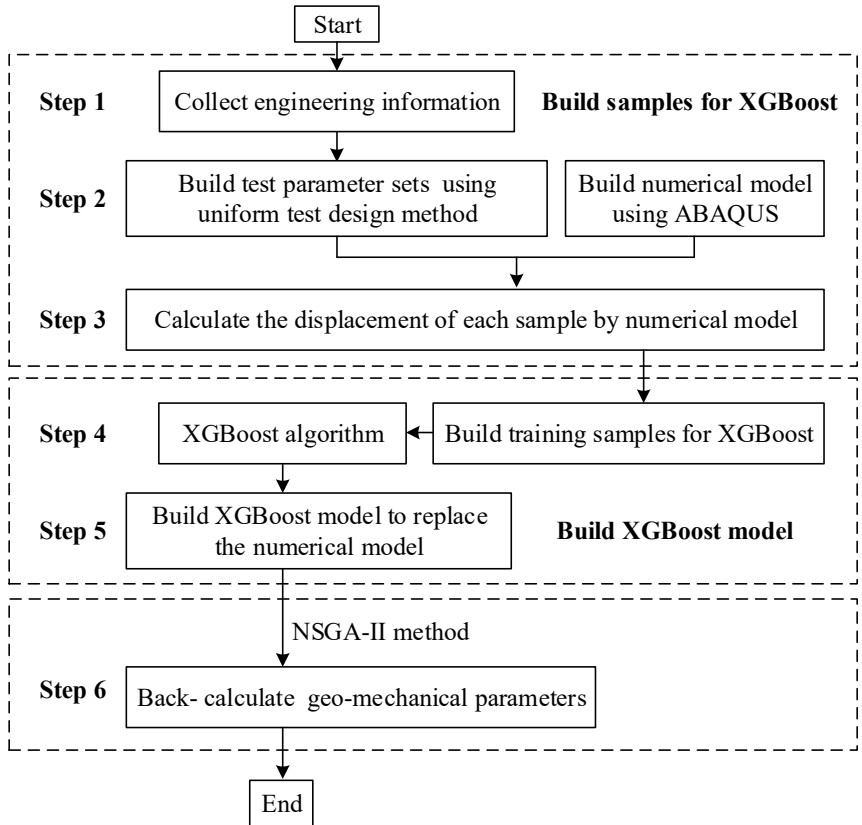

**Figure 3.** Flowchart of XGBoost–NSGA-II-based back analysis.

*Step 1* Collect the engineering information, such as geological conditions, field test and monitoring data, indoor experimental data, and in situ geostress, etc.

*Step 2* According to the information collected in Step 1, determine the range of mechanical parameters of surrounding rock that need to be recognized in the training set, and build their sample sets by the uniform test design method and a numerical model using finite element methods (FEM) or finite difference method, such as ABAQUS and ANSYS or FLAC3D.

*Step 3* Calculate the displacement values of the surrounding rock primary support system corresponding to each mechanical parameter sample.

*Step 4* Build the sample set for XGBoost. The mechanical parameter sets of the surrounding rock can be built. The displacement values of each sample set, calculated by using FEM or other numerical methods, are used to define the XGBoost model. Sample sets are composed of the geomechanical parameters of monitoring points and their corresponding displacement values.

*Step 5* Based on the sets of samples built in Step 4, the XGBoost model can be obtained by solving Equation (15).

*Step 6* Build the NSGA-II model based on Equations (16) and (17) to back-calculate the geomechanical parameters.

## 5. Case Study

### 5.1. Problem Description and Its Parameters

In the section, the Qinzhou–Sanyangchuan soft rock tunnel located in Gansu Province, People's Republic of China, is investigated as an illustrative example. Surrounding rock of Section K3 + 038~100 of the tunnel is strongly weathered granite gneiss. The buried depth of Section K3 + 038~100 of the tunnel is 135~152 m. The aggregate score of the rock mass rating (RMR) is 33, and its rock mass classification with RMR is determined to be IV. The tunnel has a height of 10.23 m and a span of 12.46 m, shown in Figure 4.

Design parameters of the initial support of the tunnel: length of the rock bolt $\phi$25-5 = 3.5 m; spacing of the grille steel frame $\phi$25 = 0.6 m; thickness of sprayed concrete = 25 cm; steel mesh $\phi$8 = 20 × 20 cm. Design parameters of the second lining of the tunnel: thickness of reinforcement = 33~45 cm. The buried depth of the tunnel is 148 m. Horizontal geostress in the area of the tunnel is 4.125~4.411 MPa. The design parameters of initial support and second lining are shown in Table 1.

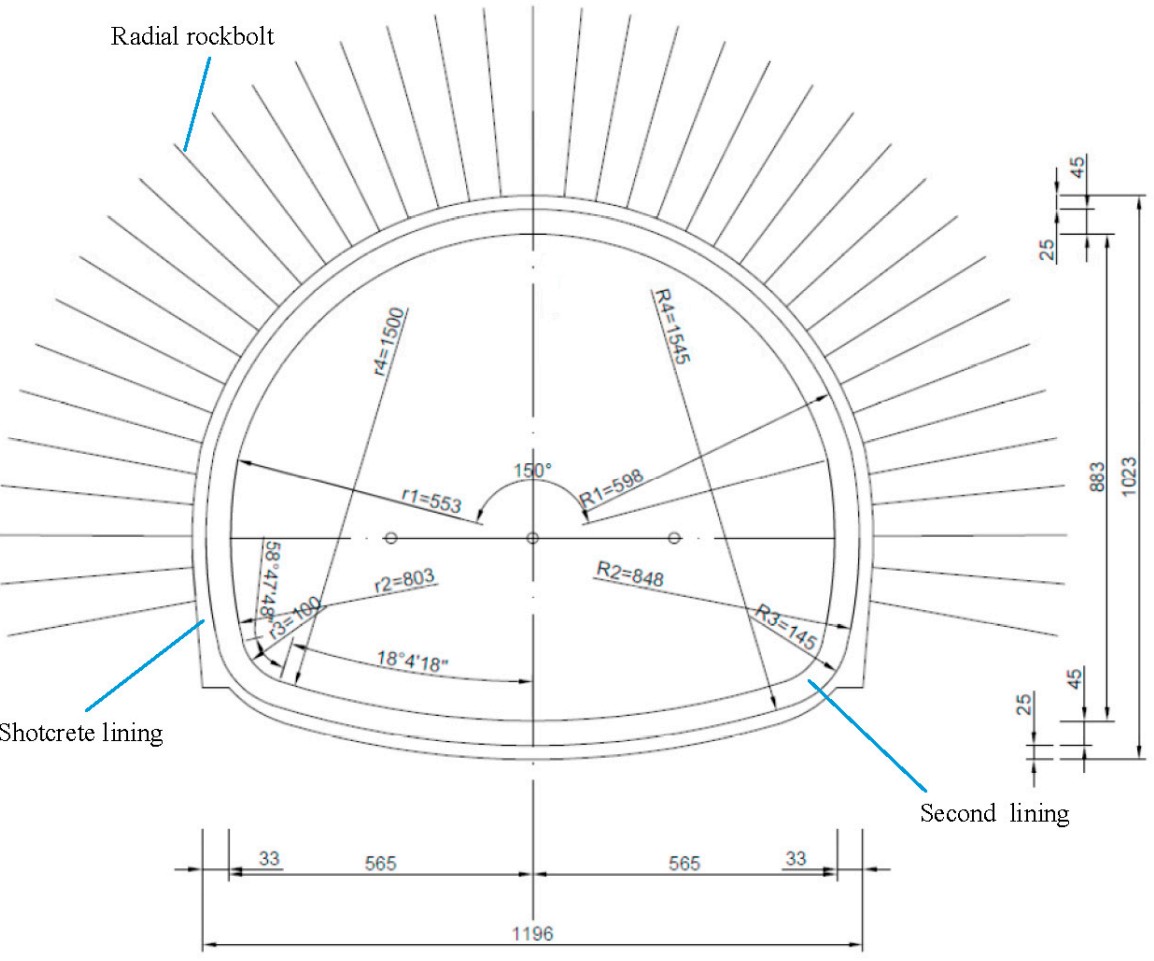

**Figure 4.** Initial support and second lining of tunnel (unit: cm).

**Table 1.** Tunnel support mechanical parameters.

| Initial Support | | Concrete Lining | | Rock Bolt | |
|---|---|---|---|---|---|
| $E_{IS}$/GPa | Poisson's Ratio $\nu$ | $E_{CL}$/MPa | Poisson's Ratio $\nu_{CL}$ | $E_{bolt}$/MPa | $\nu_{bolt}$ |
| 26.311 | 0.22 | 28,000 | 0.27 | 210,000 | 0.3 |

The initial support of Section K3 + 038~050 of the tunnel was completed on 9 October 2019. Since the completion of the initial support, the deformation of tunnel initial support structure had been increasing, leading to the collapse of Section K3 + 038~050 of this tunnel on 12 November 2019, show in Figure 5. The tunnel was monitored by the infrared rangefinder during the whole period before the disaster. On the collapse of these sections of the tunnel, the maximum value of the vault settlement and the horizontal displacement of one of side walls are 46.2 cm and 40.0 cm, respectively.

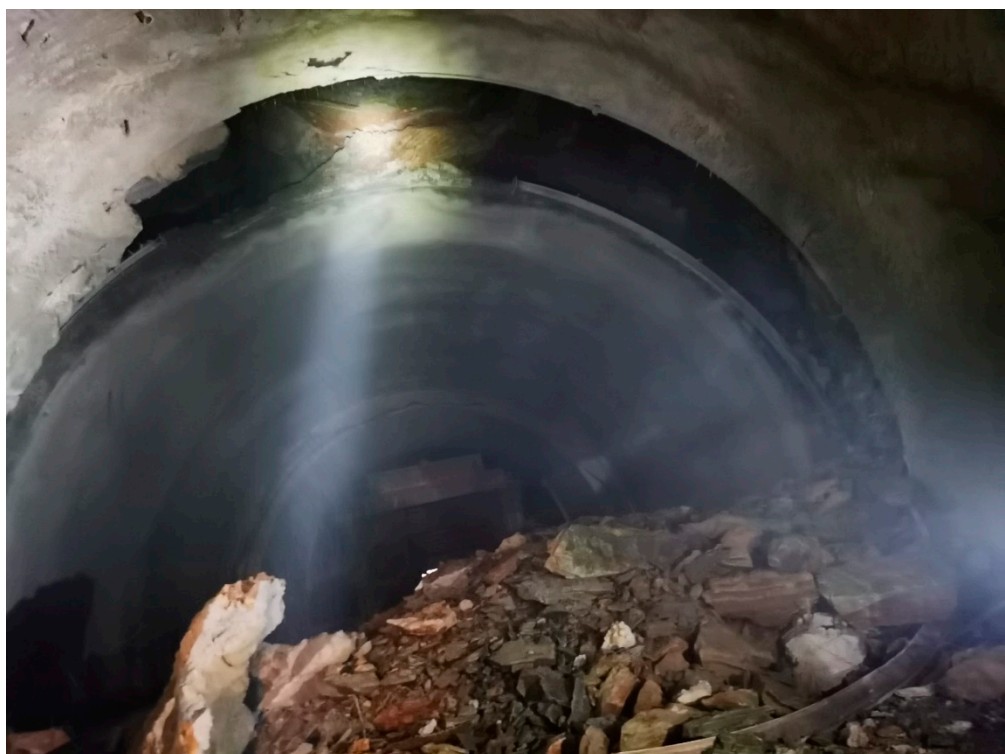

**Figure 5.** Collapse of tunnel primary support structure.

### 5.2. Modelling Approach

ABAQUS is a powerful finite element software for engineering simulation owned by Dassault SIMULIA. In this paper, we develop the NGNRM and implement it in ABAQUS_2020. The NGNRM is used to simulate the evolution process of the non-stationary parameter creep large deformation of the tunnel's surrounding rock primary support system, revealing its disaster mechanism.

### 5.3. Equivalence of Initial Support Parameters

In order to facilitate the numerical simulation, the design parameters of the tunnel's initial support structure need to be equivalent, as follows:

(1) The spacing of the grille steel frame is 0.6 m, and the initial support thickness is 0.25 m. Take the initial support of a rectangle with width $b = 1.8$ m and height $h = 0.25$ m as the model. The stiffness of the model is calculated by the equivalent stiffness method.

The equivalent concrete elastic modulus of the shotcrete supported by the grillage steel frame $E_1'$ can be calculated by Equation (20).

$$E_1' I_1 = E_1 I_1 + E_2 \sum I_2 \tag{20}$$

where $E_1$ is the elastic modulus of the concrete; $I_1$ is the moment of concrete inertia for the calculation model; $E_2$ is elastic modulus of the steel bar; $I_2$ is the moment of inertia of the virtual axis of each bar in the grille steel frame, denoted by Equation (21). In the study, $E_1 = 23$ GPa, $E_2 = 210$ GPa.

$$I_2 = \frac{\pi d^4}{64} + A b_1^2 \tag{21}$$

where $d$ is the circle diameter of the section of steel bar; $A$ is the circle area of the bar section; $b_1 =$ the half of the thickness of the initial support structure minus the thickness of the protective layer and a steel bar radius.

In this illustrative example, $E_1' = 26.311$ GPa, obtained by Equation (20).

(2) The reinforcement effect of grouting a rock bolt on surrounding rock can be equivalent to improving the cohesion of surrounding rock, which may be obtained by Equation (22).

$$c_1 = c_0 + \eta \frac{\tau_s S_s}{ab} \tag{22}$$

where $c_0$, $c_1$ are, respectively, the cohesion of surrounding rock before and after reinforcement; $\tau_s = \frac{\sigma_{bs}}{\sqrt{3}}$, $\sigma_{bs}$ is the yield strength of the rock bolt; $S_s$ is the cross-sectional area of the rock bolt; $a$ and $b$ are, respectively, longitudinal spacing and transverse spacing; $\eta$ is an experience coefficient. In this paper, $\eta = 0.963$; $\sigma_s = 335$ MPa; $a = 0.6$ m; $b = 1.2$ m.

In this illustrative example, $c_1 = c_0 + 127.05$ kPa, obtained by Equation (22).

### 5.4. Establishment of the Numerical Model

A tunnel model is established as shown in Figure 6. The ground stress was obtained from the field investigation. The size of the model is 80 m × 75 m (length × height). The displacement constraints are applied on the bottom and sides of the model, horizontal ground stress is applied on both sides of the model, and self-weight stress is applied to the entire model. Considering the symmetry of the model, we arranged six monitoring points D1~D6 on the right side of the initial support structure of the tunnel (Figure 6).

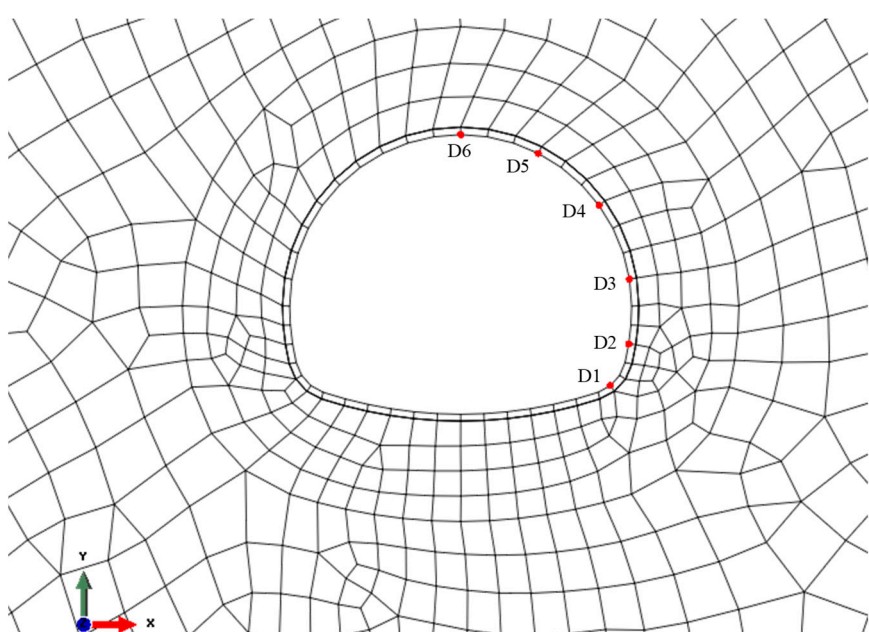

**Figure 6.** Tunnel model.

### 5.5. Determination of Numerical Simulation Parameters

We use the proposed XGBoost–NSGA-II-based back analysis method to inversely analyze the mechanical parameters of surrounding rock according to the displacement monitoring data of D3, D4 and D6 of tunnel Section K3 + 060, and obtain the mechanical parameters in Table 2. In Table 2, $\rho$ is the density of the rock mass; $E_0$ is Young's modulus of the rock mass; $\nu$ is Poisson's ratio of the rock mass; $E_1$ is the elastic modulus of a Hooke body in a Kelvin body; $\eta_1$ is the viscosity coefficient of a Newtonian body in a Kelvin body; $c_0$ is the cohesion of surrounding rock; $\varphi$ is the internal friction angle of the surrounding rock; $\lambda$ is the coefficient of confinement pressure; $\sigma_f$ and $\sigma_s$ are, respectively, the long-term strength and instantaneous strength of a Bingham body; $E_1'$ is the equivalent concrete elastic modulus of the shotcrete supported by the grillage steel frame; $A$ is a dimensionless parameter; $T$ is the time for the surrounding rock to yield.

**Table 2.** Numerical simulation parameters of the model.

| $\rho$/kg/m$^3$ | $E_0$/GPa | $\nu$ | $E_1$/GPa | $\eta_1$/GPa.h | $c_0$/MPa | $\varphi/^\circ$ |
|---|---|---|---|---|---|---|
| 2370 | 2.3 | 0.33 | 8.01 | 12,050 | 0.21 | 33 |
| $\sigma_f$/MPa | $A$ | $T$ | $\lambda$ | $\sigma_s$/MPa | $E_1'$/GPa | |
| 8.0 | 4.2 | 1650 | 1.188 | 35.62 | 26.311 | |

Then, the obtained mechanical parameters are substituted into the model for calculation. In addition, the relative errors (REs) between the calculated values and the measured values of displacements are 0.136~4.654%, 0.313~4.737%, and 0.25~3.959% for the monitoring points D3, D4, and D6, respectively. The comparison between the calculated values and the measured values of D3, D4, and D6 displacements is shown in Figure 7. Combining with Figure 7 and the REs of the displacements of each measuring point, we can note that the accuracy of the surrounding rock mechanical parameters obtained by the proposed back analysis method can meet the actual engineering requirements.

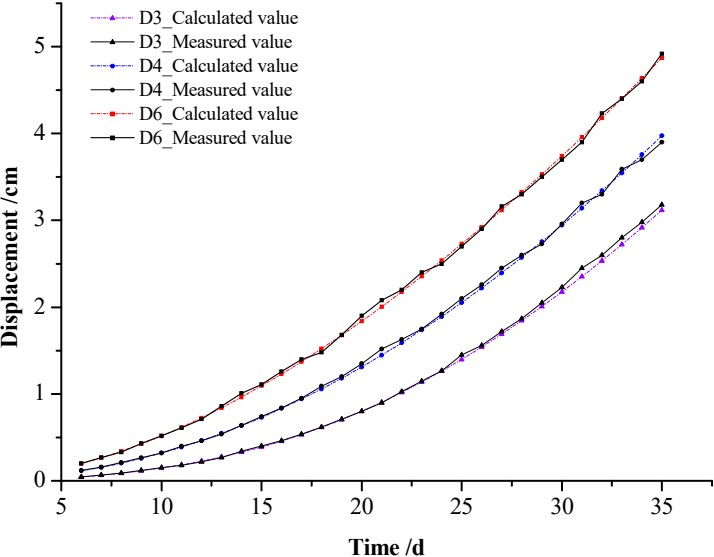

**Figure 7.** Comparison of measured and calculated values.

### 5.6. Parametric Study

In the stability analysis of the tunnel using the NGNRM, thirteen parameters of non-stationary viscoelastic–plastic model are shown in Table 2.

In this paper, the orthogonal experimental design method (OEDM) is used to study the sensitivity of nine parameters of NGNRM ($E_0$, $\nu$, $c_0$, $\varphi$, $E_1$, $\eta_1$, $\sigma_f$, $\lambda$, $E_1'$) on the vault settlement and horizontal displacement of the side wall of the tunnel. The nine parameters are selected as test factors. The values of these nine parameters in Table 2 and the values of their upper and lower 20% are taken as the level of the test factors. In addition, four empty columns are added to test the significance of the test factors. Therefore, $L_{27}(3^{13})$ is selected as the orthogonal design test table.

For vault settlement and horizontal displacement of the side wall, the variances of the nine parameters are calculated, respectively, and the statistics $F$ is constructed.

Variance of column $j$ in the orthogonal table can be expressed by Equation (23).

$$MS_j = \frac{SS_j}{df_j} \tag{23}$$

where $SS_j$ is the sum of squares of deviation; $df_j$ is the degrees of freedom (DOF) of column $j$ in the orthogonal table.

Variance of the error term in the orthogonal table may be expressed by Equation (24).

$$MS_{\text{error}} = \frac{SS_{\text{error}}}{df_{\text{error}}} \tag{24}$$

where $SS_{\text{error}}$ is the sum of squares of deviation of error term; $df_{\text{error}}$ is the sum of DOF of error term in the orthogonal table.

As thus, $F$ of factor $j$ can be calculated by Equation (25).

$$F_j = \frac{MS_j}{MS_{\text{error}}} \tag{25}$$

In order to improve the sensitivity of the $F$-test, all factors of $MS_j \leq 2MS_{\text{error}}$ together with four empty columns are included in the error term, expressed by $e^{\Delta}$; $F_{\alpha}$ is the critical value of $F$; $e$ is the error term considering only two empty columns; SOV is source of variation; Sig. denotes the significance of the influencing factor.

Finally, the significance test of the effect of these test factors is quantitatively assessed by their variances.

Based on the significance (Sig.) of the influencing factors on the vault settlement of the tunnel, we obtained the sensitivity ranking of the influencing factors when $F_j > F_{0.01(8,12)}$ as follows: $\nu > \varphi > E_0 > \lambda$. Based on the significance of the influencing factors on the sidewall horizontal displacement of the tunnel, we obtained the sensitivity ranking of the influencing factors when $F_j > F_{0.01(8,14)}$ as follows: $\nu > \varphi > \lambda > E_0$.

Therefore, $\nu$, $\varphi$, $E_0$, $\lambda$ may be determined as the main parameters of the NGNRM by the above significance analysis of the influencing factors.

### 5.7. Results and Discussion

(1) Influence of surrounding rock Poisson's ratio

From Figure 8, we note that with the increase in the reduction rate of surrounding rock Poisson's ratio, tunnel perimeter displacements increase with time, but they are only slightly changed. In general, the curvature of the curve of the vault settlement versus time (Figure 8b) is larger than that of the displacement of the side wall versus time (Figure 8a).

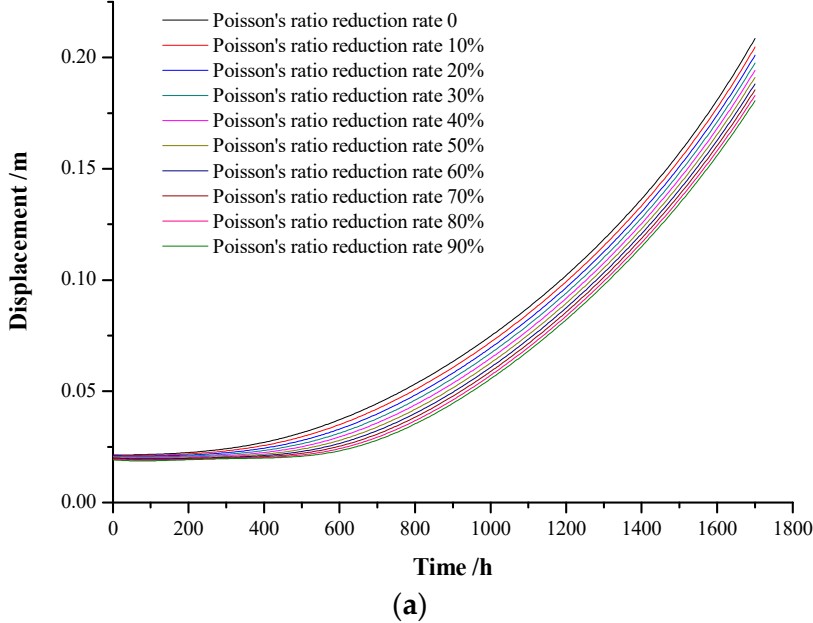

**(a)**

**Figure 8.** *Cont.*

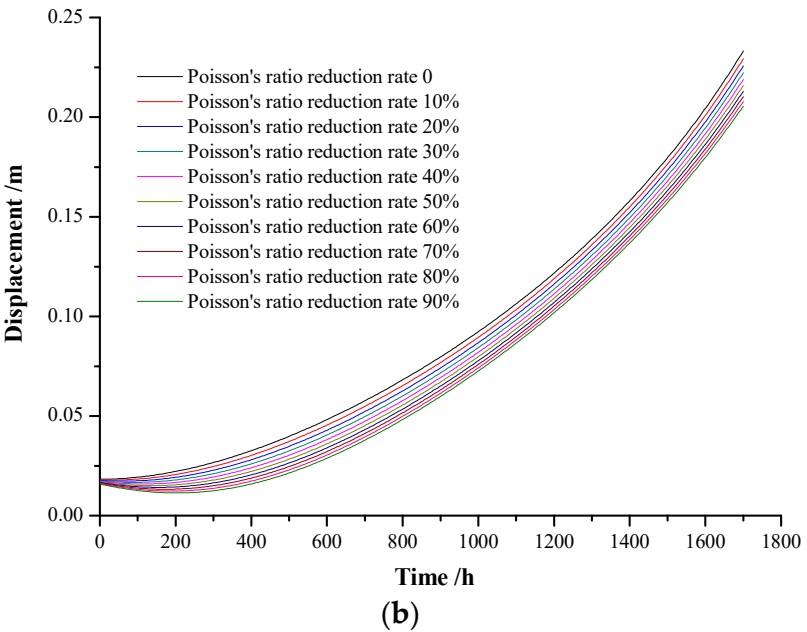

**(b)**

**Figure 8.** Variation of perimeter displacement with Poisson's ratio. (**a**) D3 monitoring point. (**b**) D6 monitoring point.

(2) Influence of surrounding rock internal friction angle

From Figure 9, we note that with the increase in the reduction rate of the surrounding rock internal friction angle, tunnel perimeter displacements increase with time, but there is a big change. With the increase in the reduction rate of the surrounding rock internal friction angle, both the curvature of the curve of the displacements of the side wall versus time (Figure 9a) and that of the vault settlement versus time (Figure 9b) decrease.

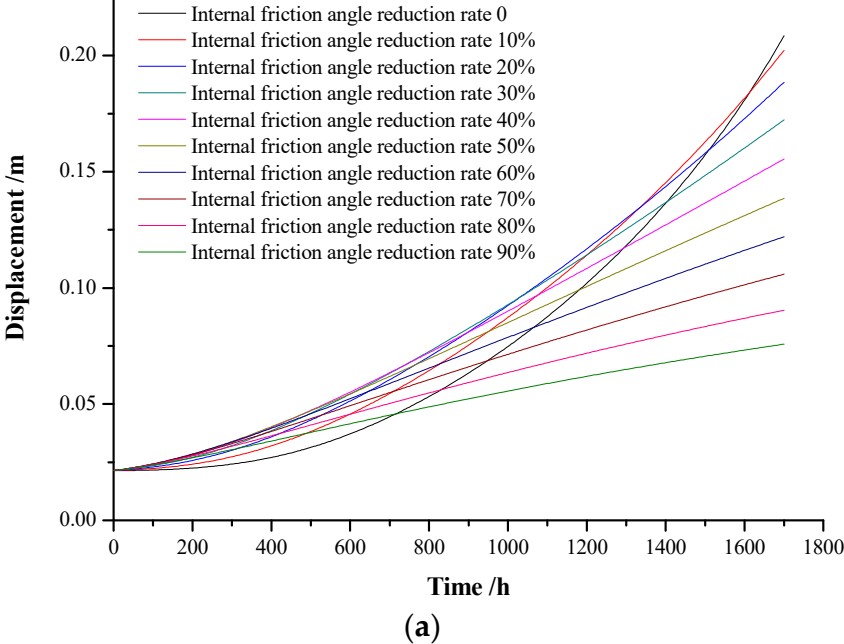

**(a)**

**Figure 9.** *Cont.*

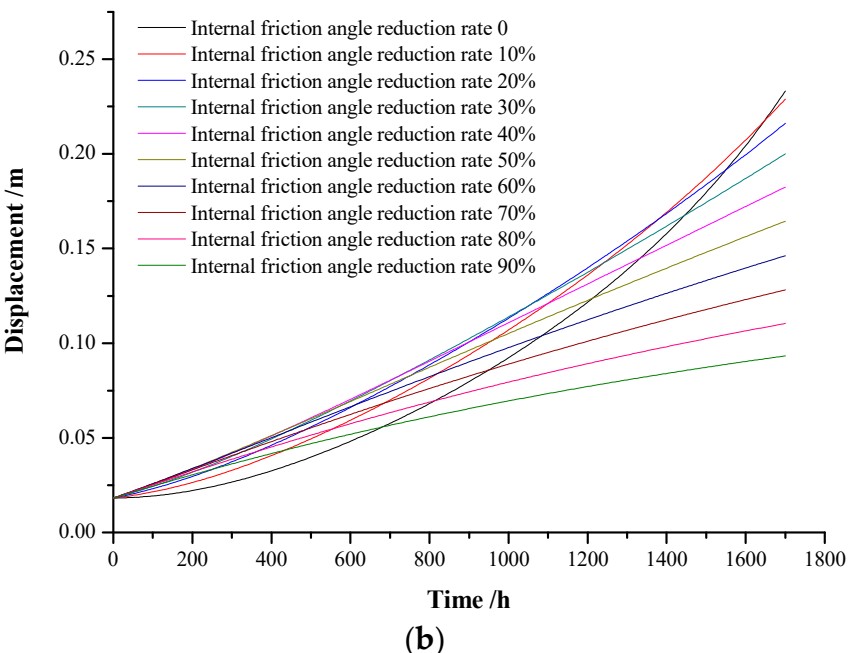

**Figure 9.** Variation of perimeter displacement with internal friction angle. (**a**) D3 monitoring point. (**b**) D6 monitoring point.

(3) Influence of the surrounding rock elastic modulus

From Figure 10, we note that with the increase in the reduction rate of the surrounding rock elastic modulus, tunnel perimeter displacements increase with time, but there is a big change. With the increase in the reduction rate of the surrounding rock elastic modulus, both the curvature of the curve of the displacements of the side wall versus time (Figure 10a) and that of the vault settlement versus time (Figure 10b) increase.

(4) Influence of the lateral pressure coefficient

From Figure 11, we note that with the increase in the reduction rate of the lateral pressure coefficient, tunnel perimeter displacements increase with time. However, for the reduction rate of 10% among the lateral pressure coefficient, the curvature of the curve of the perimeter displacements and lateral pressure coefficient increases with time and its variation is the largest. In general, for the reduction rate of 10~30% among the lateral pressure coefficient, the curvature of the curve of the perimeter displacement and lateral pressure coefficient increases with time. The curvature of the curve corresponding to the other reduction rates are only slightly changed.

(5) Influence of the elastic modulus of primary support

From Figure 12, we note that with the increase in the reduction rate of the elastic modulus of primary support, tunnel perimeter displacements increase with time. In general, the curvature variation of the curve of the displacements of the side wall versus time is only slightly changed (Figure 12a). However, with the increase in the reduction rate of the elastic modulus of primary support, the curvature of the curve of the vault settlement versus time (Figure 12b) has a relatively large change. That is to say, the initial support stiffness has a greater influence on the vault settlement than the side wall displacement.

(6) Non-stationary parameter creep large deformation mechanism

In the following, we use the proposed method to investigate the large deformation mechanism of Section K3 + 038~040 of the tunnel.

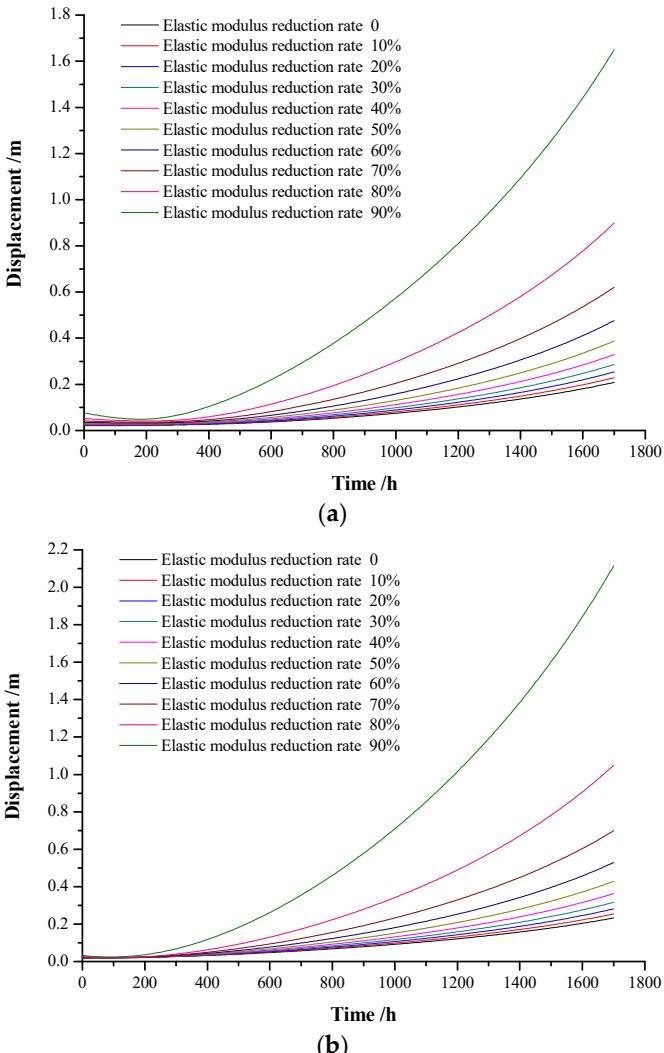

**Figure 10.** Variation of perimeter displacement with elastic modulus. (**a**) D3 monitoring point. (**b**) D6 monitoring point.

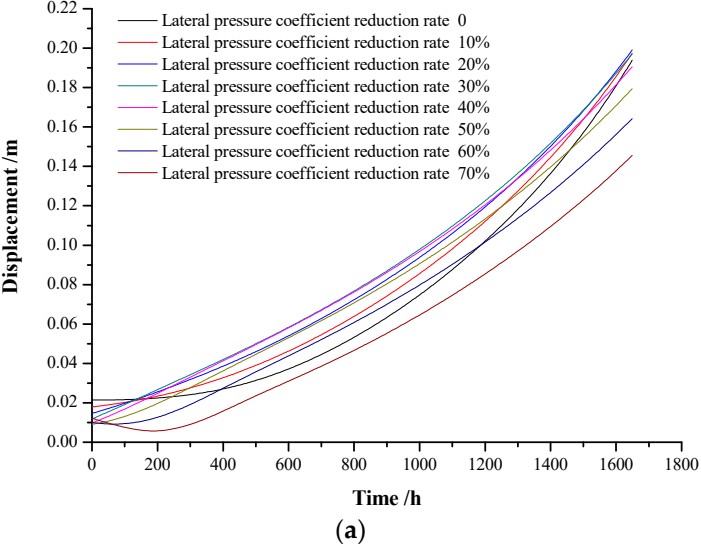

**Figure 11.** *Cont*.

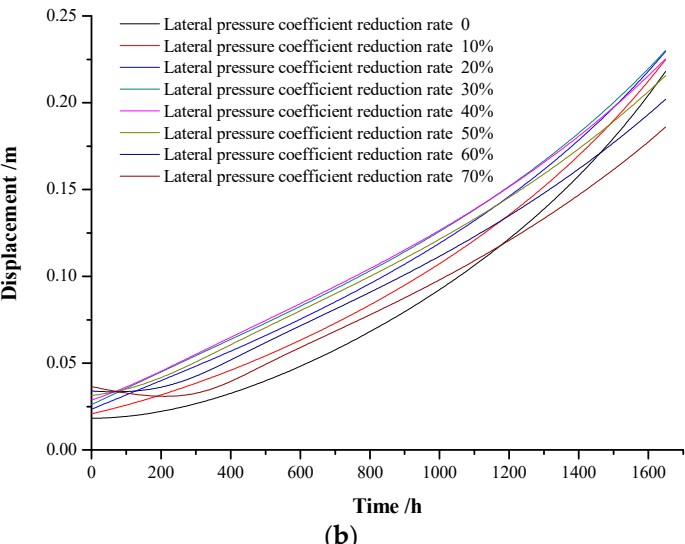

**Figure 11.** Variation of perimeter displacement with lateral pressure coefficient. (**a**) D3 monitoring point. (**b**) D6 monitoring point.

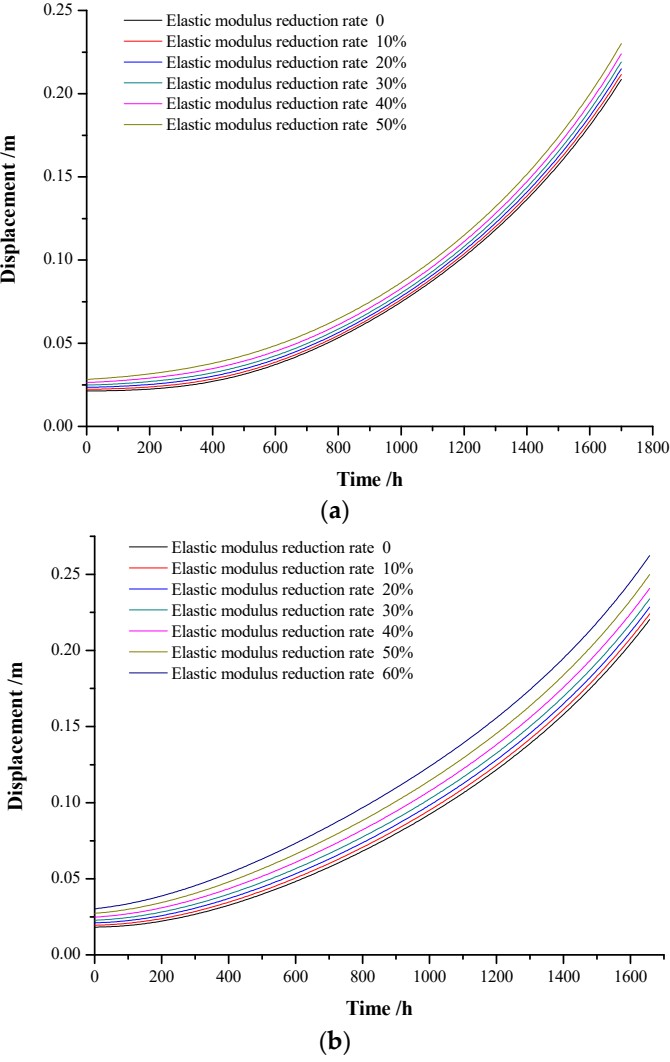

**Figure 12.** Variation of perimeter displacement with elastic modulus of primary support. (**a**) D3 monitoring point. (**b**) D6 monitoring point.

We use the proposed back analysis method to inversely analyze the mechanical parameters of surrounding rock according to the displacement monitoring data of monitoring points D4 and D6 of Section K3 + 040 from 1 to 24 h of rheology time. Then, the obtained mechanical parameters are substituted into the model for calculation. In addition, the REs between the calculated values and the measured values of displacements are 0.116~4.661 and 0.19~4.015 for the monitoring points D4 and D6, respectively. The comparison between the calculated values and the measured values of D4 and D6 displacements is shown in Figure 13. Combining with Figure 13 and the REs of the displacements of each measuring point, we can note that the accuracy of the surrounding rock mechanical parameters obtained by the proposed back analysis method can meet the actual engineering requirements.

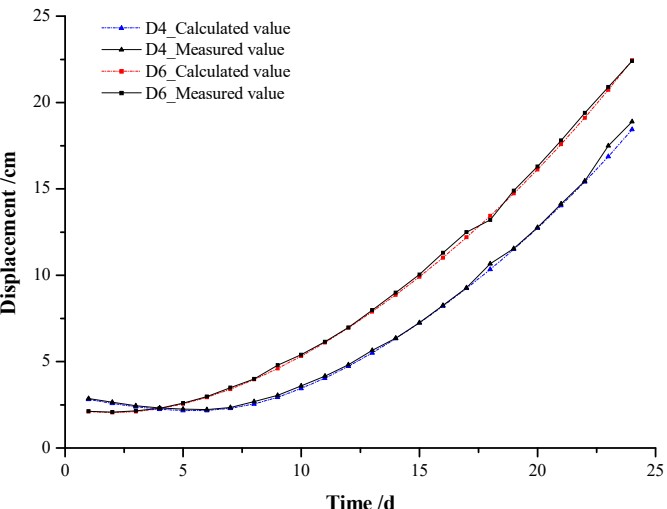

**Figure 13.** Variation of perimeter displacement with elastic modulus of primary support.

From Figures 14–16, we note that the displacements of monitoring points No. 1 (Node 8), No. 2 (Node 69), No. 3 (Node 13), No. 4 (Node 70), No. 5 (Node 17), and No. 6 (Node 20) on tunnel primary support structure present an approximate upward concave parabolic change with rheological time. The displacements of the monitoring point No. 1 at the vault have the largest among them.

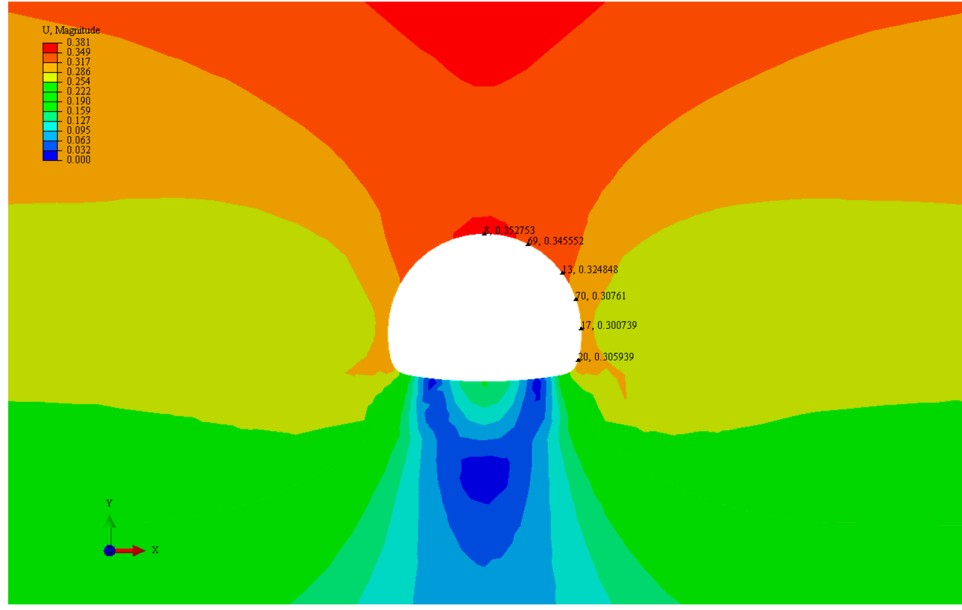

**Figure 14.** Contour map of tunnel displacement at rheological time 721 h.

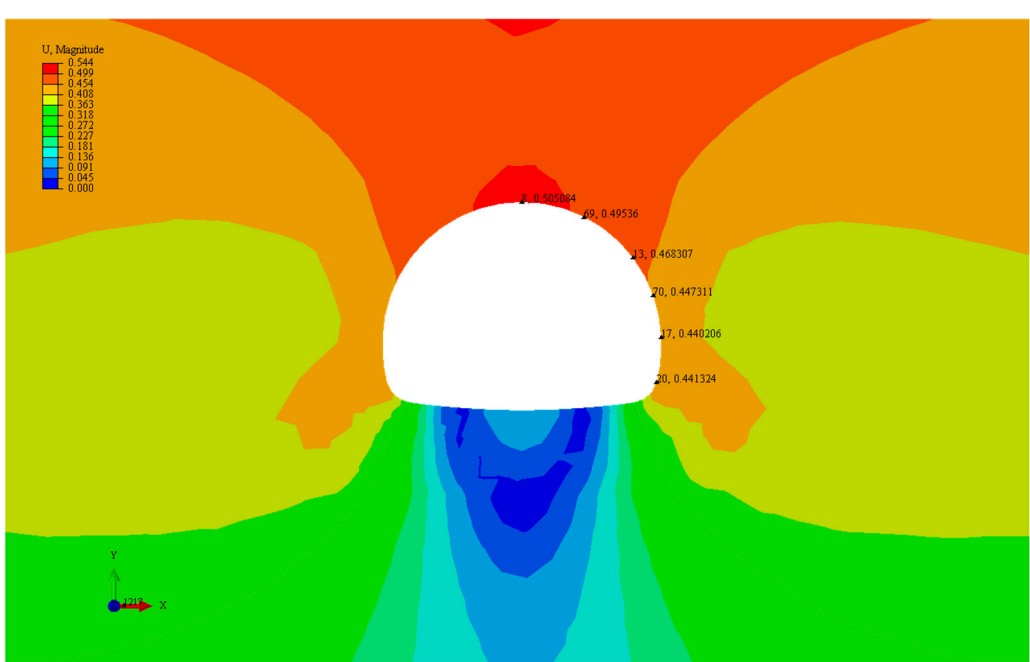

**Figure 15.** Contour map of tunnel displacement at rheological time 837 h.

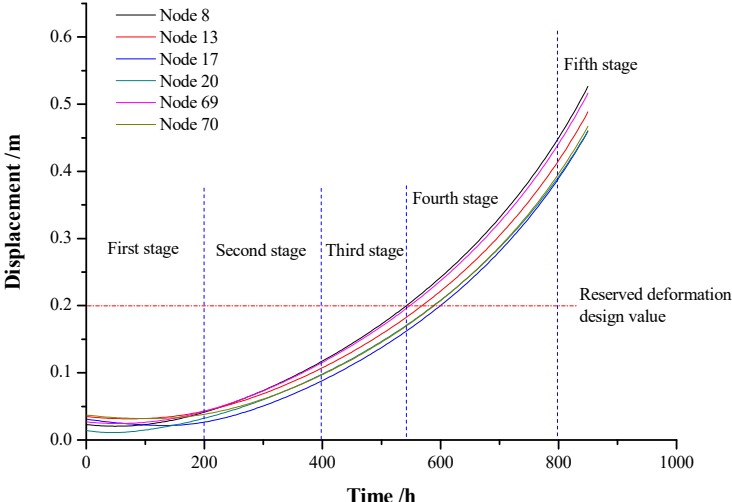

**Figure 16.** Variation curve of the displacements of tunnel measuring points with time.

From Figures 14–16, and Table 3, we note that the displacements of the monitoring points from the vault to the side wall basically present a sequence from large to small, which is consistent with the collapse of the tunnel primary support structure shown in Figure 5.

**Table 3.** Displacement and velocity of monitoring points around the tunnel.

| Monitoring Point Number | Displacement/m | | Displacement Average Rate/cm/d |
|---|---|---|---|
| | Time 30 d | Time 34.833 d | |
| NO.1 | 0.353 | 0.505 | 3.145 |
| NO.2 | 0.346 | 0.495 | 3.083 |
| NO.3 | 0.325 | 0.468 | 2.959 |
| NO.4 | 0.308 | 0.447 | 2.876 |
| NO.5 | 0.301 | 0.440 | 2.876 |
| NO.6 | 0.306 | 0.441 | 2.793 |

From Figure 16, we also note that the large deformation of the initial support of the tunnel has undergone the following five stages of evolution:

(1) At the first stage, the displacement rate is very small, for example, its average displacement rate of monitoring point NO.1 is 0.226 cm/d from 0 h to 200 h of rheology time.

(2) The average displacement rate at the second stage is much larger than that at the first stage. For example, the average displacement rate of monitoring point NO.1 at the second stage is 0.915 cm/d from 200 h to 400 h of rheology time.

(3) The average displacement rate at the third stage is much larger than that at the second stage. For example, the average displacement rate of monitoring point NO.1 at the third stage is 1.390 cm/d from 400 h to 542.5 h of rheology time. The displacement of the primary support structure of the tunnel reaches the reserved deformation design value 20 cm when the surrounding rock force rheology time is 542.5 h.

(4) The average displacement rate at the fourth stage is very much larger than that at the third stage. For example, the average displacement rate of monitoring point NO.1 at the fourth stage is 2.336 cm/d from 542.5 h to 800 h of rheology time.

(5) The average displacement rate at the fifth stage is very much larger than that at the fourth stage. For example, the average displacement rate of monitoring point NO.1 at the fifth stage is 3.695 cm/d from 800 h to 850 h of rheology time.

From Figure 17, we also note that the displacement velocities of the monitoring points increase approximately linearly from the sixth day to the twentieth day after rheology begins, and then it increases rapidly, especially after 28 days of rheology. That is to say, it is most scientific and reasonable to reinforce and dispose of the large deformation disaster of the tunnel within 20 days after the initial support.

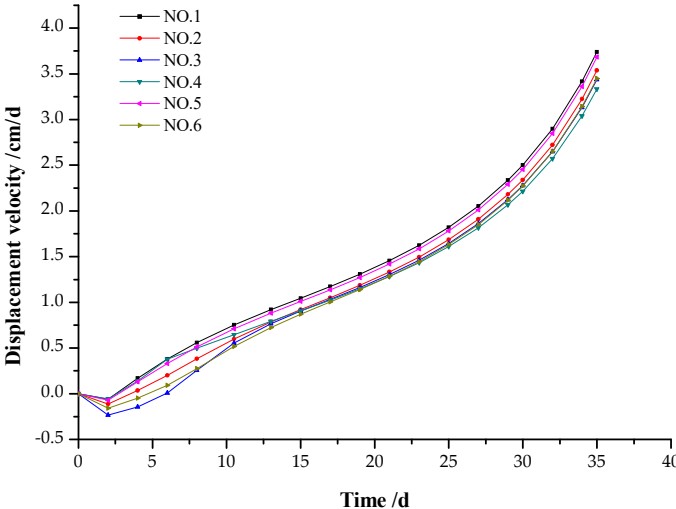

**Figure 17.** Curve of displacement velocities and time of measuring points.

## 6. Conclusions

1. The NGNRM with non-stationary parameter creep developed in this paper can effectively evaluate the large deformation mechanism of soft rock tunnels.

2. The proposed XGBoost–NSGA-II-based back analysis can simultaneously obtain multiple mechanical parameters of the surrounding rock using machine learning algorithms to construct surrogate models of numerical simulation, and optimize parameters of the surrogate model.

3. Through the analysis of the significance of the influencing factors, it is shown that $v$, $\varphi$, $E_0$, and $\lambda$ are four main parameters of the NGNRM affecting the large deformation of soft rock tunnels.

4. The large deformation of the initial support of the tunnel generally undergoes the five stages of evolution from a very small to a very high displacement rate.

5. For soft rock tunnels, the displacement velocities of monitoring points increase approximately linearly within a certain period of time after rheology begins, and then it increases rapidly, especially with large acceleration, in a very short period of time before the collapse of the surrounding rock primary support system.

6. It is most scientific and reasonable to reinforce and dispose of the large deformation disaster of a tunnel before the displacement rate of monitoring points changes from uniform variation to accelerated development.

**Author Contributions:** J.X.: conceptualization and constitutive model development; H.W.: numerical simulation and analysis; C.S.: Back analysis method development; C.Y.: software; writing—review and editing; G.R.: data curation, investigation. All authors have read and agreed to the published version of the manuscript.

**Funding:** This research was funded by Science and Technology Project of China Railway 20th Bureau Group Co., Ltd., and China Postdoctoral Science Foundation, grant number 20060390165.

**Institutional Review Board Statement:** Not applicable.

**Informed Consent Statement:** Not applicable.

**Data Availability Statement:** Data are contained within this article.

**Conflicts of Interest:** The authors declare no conflict of interest.

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
