# Peer review of "Numerical Simulation of Non-Stationary Parameter Creep Large Deformation Mechanism of Deep Soft Rock Tunnel"

_applsci, doi:10.3390/app12115311_

Round 1

Reviewer 1 Report

The employed method to identify the crucial rheological parameters of soft rocks should provide the best fit of predicted to actual (MONITORED) displacement sets. Therefore, it would be highly desirable for the reader to understand in more detail the monitoring routine. Whether it was performed on the considered tunnel during the whole period before the disaster, or another approach was used? Such a clear explanation is wanted in the introduction or/and Section 5.

Author Response

(1) According to the suggestions of reviewers, we had provided the best fit of predicted to actual displacement sets, and added the analysis of the relative errors (REs) between the calculated values and the measured values of dis-placements in Section 5 of the revised manuscript.

(2) The considered of tunnel was monitored by the infrared rangefinder during the whole period before the disaster. According to the displacement data of the completed sections or a period of time of the inversion section, we used the proposed XGBoost – NSGA-II based MBAM to identify the crucial rheological parameters of soft rocks. We had made the clear explanation in the introduction and Section 5 of the revised manuscript. 

Reviewer 2 Report

In this paper, the authors propose machine learning tools to help in the understanding of numerical simulation rheological models in large creep evolution during soft rock tunneling.  XGBoost, an extreme gradient boosting algorithm is used to evaluate the loss function and the non-dominated sorting genetic algorithm II to optimize it. The consistency of the models with a confirmed case of tunnel failure is studied.

The paper is well written and correctly structured, and it is clear about its significance. In my opinion, only minor changes are required, as follows:

  1. Explain what UMAT means in the text and the flowchart in figure 2. Moreover, provide a more detailed description of this figure
  2. Please justify why Pc=0.4-0.9 was proposed as the probability of generating new chromosomes in step 6 of the genetic algorithm. It is a pretty broad range.
  3. Correct the font format to steps 1 – 3 in section 4.3.
  4. Improve tables 1, 2, and 3, to correctly fit the page margins.
  5. Correct the form of indicating the units of the axes in the figures: 7, 8, 9, 10, 11, 12, 15, and table 3.
  6. Labels inside figures 13 and 14 are not clear because of the color of the font; you should change the labels to black instead of white.
  7. In figure 15, why is only the measuring point 1 depicted?
  8. Revise some minor typos.

Author Response

  1. Explain what UMAT means in the text and the flowchart in figure 2. Moreover, provide a more detailed description of this figure.

Re:

According to the suggestions of reviewers, we had explained what UMAT means (User-defined material mechanical behavior —— UMAT is a Fortran program interface provided by ABAQUS to users to define their own material properties), and added the detailed description of Figure 2 in the revised manuscript.

  1. Please justify why Pc=0.4-0.9 was proposed as the probability of generating new chromosomes in step 6 of the genetic algorithm. It is a pretty broad range.

Re:

The crossover operation is performed by randomly selecting two individuals from the pairing library according to a certain crossover probability. The crossover location is also randomly determined. The mating probability of each chromosome is determined by the mating probability, and the crossover probability  generally ranges from 0.4 to 0.9 so as to facilitate the random selection of mating locations.

  1. Correct the font format to steps 1 – 3 in section 4.3.

Re:

According to the suggestions of reviewers, we had corrected the font format to steps 1 – 3 in section 4.3 in the manuscript.

  1. Improve tables 1, 2, and 3, to correctly fit the page margins.

Re:

According to the suggestions of reviewers, we had improved tables 1, 2, and 3 to fit the page margins in the revised manuscript.

  1. Correct the form of indicating the units of the axes in the figures: 7, 8, 9, 10, 11, 12, 15, and table 3.

Re:

According to the suggestions of reviewers, we had corrected the form of indicating the units of the axes in the figures: 8, 9, 10, 11, 12, 13, 16, 17, and table 3. in the revised manuscript.

  1. Labels inside figures 13 and 14 are not clear because of the color of the font; you should change the labels to black instead of white.

Re:

According to the suggestions of reviewers, we had change the labels inside figures 14 and 15 to black.

  1. In figure 15, why is only the measuring point 1 depicted?

Re:

According to the suggestions of reviewers, we had supplemented the data for all 6 measuring points in Figure 17 of the revised manuscript.

Reviewer 3 Report

It is an interesting topic, but some questions should be answered:

  • Highlight the novelty of the research. It is a very detailed and well-structured manuscript, but there is a lack of novelty regarding research that should be boosted in order to be published.
  • More characteristics of the case study are needed, for instance: depth, rock mass classification with RMR, Q, RMi or similar.
  • Version of the software used
  • The format must be kept, for instance the case of equations or some paragraphs like 287-286.

Author Response

(1) According to the suggestions of reviewers, we had highlight the novelty of the research as follows in the revised manuscript. The novelty of this paper is that we propose the NGNRM (a nonlinear generalized Nishihara rheological model with the non-stationary-parameter-creep) to fully express the acceler-ated creep characteristics of soft rock under high stress, and the XGBoost – NSGA-II based multi-objective back analysis method, to help in the understanding of numerical simulation rheological models in large creep evolution during soft rock tunneling.

(2) According to the suggestions of reviewers, we had supplemented some characteristics of the case study such as depth and rock mass classification with RMR in the revised manuscript. The buried depth of the tunnel is 148 m. The aggregate score of rock mass rating (RMR) is 33, and its rock mass classification with RMR is determined to be IV.

(3) According to the suggestions of reviewers, we had corrected the inconsistency format of equations or some paragraphs like 277-286, and kept the consistent format in the revised manuscript.
